## Sex-specific associations between surgery-induced weight loss and cancer outcomes: A post hoc analysis of the prospective, controlled Swedish Obese Subjects study

Kajsa Sjöholm[1], Per-Arne Svensson[1,2], Johanna C. Andersson-Assarsson[1], Peter Jacobson[1], Sofie Ahlin[1,3], Cecilia Karlsson[1,4], Björn Carlsson[1,5], Felipe M. Kristensson[1,6], Per Karlsson[7], Markku Peltonen[8], Lena M. S. Carlsson[1], Magdalena Taube[1]*

1 Institute of Medicine, Sahlgrenska Academy at the University of Gothenburg, Gothenburg, Sweden, 2 Institute of Health and Care Sciences, Sahlgrenska Academy at the University of Gothenburg, Gothenburg, Sweden, 3 Department of Clinical Physiology, Region of Västra Götaland, NU Hospital Group, Trollhättan, Sweden, 4 Late-Stage Development, Cardiovascular, Renal and Metabolism (CVRM), BioPharmaceuticals R&D, AstraZeneca, Gothenburg, Sweden, 5 Research and Early Development, Cardiovascular, Renal and Metabolism (CVRM), BioPharmaceuticals R&D, AstraZeneca, Gothenburg, Sweden, 6 Department of Surgery, Region Västra Götaland, Sahlgrenska University Hospital/Östra, Gothenburg, Sweden, 7 Department of Oncology, Institute of Clinical Sciences, Sahlgrenska Academy, University of Gothenburg, Gothenburg, Sweden, 8 Finnish Institute for Health and Welfare, Helsinki, Finland

* magdalena.taube@wlab.gu.se

## Abstract

### Background

Obesity increases cancer risk, whereas surgery-induced weight loss is associated with reduced risk. Risk-based patient stratification may be needed to better understand and maximize benefits of weight loss interventions in individuals with obesity. To this end, comprehensive data from high-quality studies with extended follow-up are imperative. This study examines the link between bariatric surgery and long-term cancer outcomes, focusing on patient subgroups defined by previously suggested predictors of treatment benefit, such as sex and baseline insulin levels.

### Methods and findings

This post-hoc analysis used data from the Swedish Obese Subjects (SOS) study, a prospective, controlled intervention trial, designed to investigate the long-term effects of bariatric surgery-induced weight loss (ClinicalTrials.gov, NCT01479452). The study was conducted at 25 public surgical departments and 480 primary healthcare centers across Sweden. Between Sept 1, 1987, and Jan 31, 2001, 2,007 per-protocol patients with obesity who underwent bariatric surgery (gastric bypass, $n=266$; gastric banding, $n=376$; vertical banded gastroplasty, $n=1,365$) and 2,040 matched controls, receiving standard

which permits unrestricted use, distribution, and reproduction in any medium, provided the original author and source are credited.

**Data availability statement:** The data is subject to legal restrictions according to national legislation. Confidentiality regarding personal information in studies is regulated in the Public Access to Information and Secrecy Act (Offentlighets- och sekretesslagen (OSL); SFS 2009:400). There is a possibility to apply to get access to public documents that an authority holds. In this case, the University of Gothenburg is the specific authority that holds the documents. A request to get access to public documents can be rejected or granted with reservations. If the authority refuses to disclose the documents the applicant is entitled to get a written decision that can be appealed to the administrative court of appeal. Contact for data inquiries: registrator@gu.se.

**Funding:** This project was supported by grants from the Swedish Research Council (2021-01496 (LMSC) and 2020-01303 (PAS)), the Swedish state under the agreement between the Swedish government and the county councils, the ALF-agreement (ALFGBG-1005957 (KS), ALFGBG-1006165 (LMSC), ALFGBG-1006182 (PAS), ALFGBG-1006404 (MT)), the Health & Medical Care Committee of the Region Västra Götaland (VGFOUREG-931560 (SA), VGFOUREG-941125 (SA), VGFOUFBD-100464 (SA)), the Swedish Heart and Lung Foundation (20240397; LMSC), the Wilhelm and Martina Lundgren Foundation (2023-SA-4298; SA), the Royal Society of Arts and Sciences in Gothenburg (2024-867; MT), and the Adlerbert Research Foundation (AF2024-0075; KS). The funders of the study had no role in the study design, data collection or analysis, decision to publish, or preparation of the manuscript.

**Competing interests:** I have read the journal's policy and the authors of this manuscript have the following competing interests: B.C. and C.K. are employed by AstraZeneca and hold stocks in the same company. P. Karlsson reports advisory roles and lecturing for AstraZeneca, Novartis, Roche and Seagen, as well as pending patents for Exact Sciences and PreludeDx, all outside the current study. No other conflict of interest was reported. K.S. has received speaker honoraria from AstraZeneca and Encore Medical Education, all outside the current study. M.T. holds stocks in Zatisfied Health AB,

nonsurgical obesity-related care, were recruited. Inclusion criteria were age 37–60 years and a body mass index (BMI) ≥34 kg/m² for men and ≥38 kg/m² for women. The primary outcome measures were cancer events and cancer-related deaths, captured through nearly complete data sourced from national Swedish health registries. Female-specific cancers were defined as gynecologic and breast cancers. Analyses were adjusted (adj) for baseline age, sagittal diameter, alcohol consumption, smoking, and serum insulin levels. The study was closed on December 31, 2022. Median follow-up was 26.8 years (interquartile range (IQR) [22.9, 29.6]) in the surgery group and 24.9 years (IQR [18.7, 28.8]) in the control group. Bariatric surgery was associated with a lower overall cancer incidence rate in women (adjusted hazard ratio (HRadj) = 0.78; 95% confidence interval (CI) [0.67, 0.90]; $p = 0.001$), but not in men (sex–treatment interaction $p = 0.013$). The HRadj for overall cancer mortality rate in women was 0.78 (95% CI [0.61, 1.00]; $p = 0.050$). In women, surgery was associated with a lower incidence rate of both obesity-related cancers (HRadj = 0.70; 95% CI [0.58, 0.85]; $p < 0.001$) and female-specific cancers (HRadj = 0.60; 95% CI [0.47, 0.75]; $p < 0.001$). Importantly, subgroup analyses showed that the associations between surgery and female-specific cancer incidence, as well as female-specific cancer-related mortality, were stronger in women with high baseline insulin levels (insulin-treatment interaction $p = 0.021$ and 0.039, respectively). The main limitation is that cancer was not a predefined study outcome.

## Conclusions

Bariatric surgery is associated with a lower risk of cancer and cancer-related mortality in women with obesity, with the strongest association observed for female-specific cancers in women with elevated baseline insulin levels. In men, bariatric surgery was not associated with overall cancer incidence or mortality. These findings support incorporating risk-based stratification to better tailor cancer prevention strategies in obesity care.

### Author summary

#### Why was this study done?

- Obesity is a well-established risk factor for cancer, and bariatric surgery has been linked to reduced cancer incidence and mortality.

- Prior studies suggest that treatment outcomes differ by sex and metabolic status, but long-term prospective evidence on subgroup-specific cancer risk reduction is limited.

#### What did the researchers do and find?

- In this post-hoc analysis of the Swedish Obese Subjects (SOS) study, 2,007 patients who underwent bariatric surgery were compared to 2,040 matched controls over a median follow-up of 26.8 and 24.9 years, respectively.

outside the current study. No other conflict of interest was reported.

**Abbreviations:** CI, confidence interval; BMI, body mass index; HR, hazard ratio; HRadj, adjusted hazard ratio; IARC, International Agency for Research on Cancer; IQR, interquartile range; IR, incidence rate; LABS, Longitudinal Assessment of Bariatric Surgery; RMST, Restricted Mean Survival Time; SOS, Swedish Obese Subjects; STROBE, Strengthening the Reporting of Observational Studies in Epidemiology.

- Bariatric surgery was associated with lower cancer incidence and mortality rates in women, particularly for obesity-related and female-specific cancers (e.g., gynecologic and breast cancers).

- Subgroup analyses revealed that these associations were strongest for female-specific cancers in women with high baseline insulin levels.

## What do these findings mean?

- Bariatric surgery may offer long-term cancer protection for women with obesity, especially those with elevated insulin levels, highlighting the importance of metabolic profiling in treatment planning.

- These results support the use of risk-based stratification to optimize cancer prevention strategies in obesity care.

- Further research is needed to validate these findings and explore their relevance to current bariatric surgery techniques and emerging pharmacological weight-loss interventions.

- The main limitation is that cancer was not a predefined outcome when the study began.

## Introduction

Global cancer incidence is projected to rise by 77% by 2050 [1]. Obesity, a known cancer risk factor, contributes to this increase, in both high-income and low- to middle-income countries [2]. Moreover, obesity is associated with worse prognosis and higher overall cancer-related mortality [3]. The International Agency for Research on Cancer (IARC) has linked obesity to increased risk of 13 cancers, including common female cancers like postmenopausal breast and endometrial cancer [4]. The obesity epidemic is also driving a rise in obesity-related cancers among younger age groups [5]. Previous studies show that surgery-induced weight loss is linked to reduction of overall cancer risk and cancer-related mortality [6–10]. However, less is known about the effect of weight loss on cancer subtypes, and which patient subgroups benefit the most. Bariatric surgery offers a distinct opportunity to investigate the impact of substantial, long-term weight loss on cancer risk, potentially guiding future obesity-related cancer prevention strategies and the clinical use of weight-loss medications now emerging in cancer prevention research [11].

In 2009, the Swedish Obese Subjects (SOS) study became the first prospective study to report an association between surgery-induced weight loss and lower cancer incidence in women, but not men. Although the observed sex difference in treatment effect was not supported by sufficient statistical evidence at the time [10], subsequent retrospective studies have consistently suggested that women may experience more favorable outcomes [6,9,12]. Moreover, metabolic changes following bariatric surgery were recently proposed as independent predictors of reduced cancer risk, with glycemic biomarkers showing the strongest associations [13]. This is in line with previous

sub-analyses of the SOS cohort which have indicated that insulin may be a modifier of bariatric surgery benefit [14,15]. In addition, several retrospective or uncontrolled studies have indicated that the magnitude of weight loss itself may play a critical role in cancer risk reduction [7,13,16].

With 17 additional years of follow-up in the SOS study, the aim was to further examine the link between surgery-induced weight loss and long-term cancer outcomes, focusing on patient subgroups defined by previously identified predictors to optimize treatment benefit.

## Methods

### The SOS study

The SOS study, conducted at 25 surgical departments and 480 primary healthcare centers in Sweden, is a prospective matched intervention trial comparing bariatric surgery with conventional obesity treatment [8]. In brief, the study enrolled 2,010 participants who chose bariatric surgery, and 2,037 matched controls recruited between September 1, 1987, and January 31, 2001. Of the 2,010 surgery participants, 2,007 actually underwent bariatric surgery (per-protocol), while three did not undergo surgery and were therefore included in the control group in per-protocol analyses (Fig A in S1 Appendix).

Matching of the surgery and control groups was performed at the group level according to the method of sequential treatment assignment [17], based on the following matching variables: sex, age, smoking, diabetes, weight, height, waist circumference, hip circumference, systolic blood pressure, triglycerides, total cholesterol, menopausal status (pre/post), current health, monotony avoidance, psychasthenia, quantity of social support, quality of social support and stressful life events [8]. The surgery and control groups had identical inclusion and exclusion criteria and all participants in both groups were eligible for surgery. Inclusion criteria were age 37–60 years and body mass index (BMI) ≥34 kg/m$^2$ for men or ≥38 kg/m$^2$ for women. Exclusion criteria were earlier surgery for gastric or duodenal ulcer, earlier bariatric surgery, gastric ulcer in the past 6 months, ongoing malignancy, active malignancy in the past 5 years, myocardial infarction in the past 6 months, bulimic eating pattern, drug or alcohol abuse, psychiatric or cooperative problems contraindicating bariatric surgery, other contraindicating conditions (such as chronic glucocorticoid or anti-inflammatory treatment).

The intervention study began on the day of surgery for both the surgically treated individual and the matched control. Participants who had surgery underwent gastric banding (376/2,007; 19%), vertical banded gastroplasty (1,365/2,007; 68%), or gastric bypass (266/2,007; 13%), while controls received the standard nonsurgical obesity-related care provided at their primary healthcare centers, which was not standardized and ranged from structured lifestyle and behavioral interventions to minimal or no active treatment (Fig A in S1 Appendix). Both groups underwent a baseline examination approximately four weeks before the intervention. Follow-up visits, including physical examinations and questionnaires, were scheduled at 0.5, 1, 2, 3, 4, 6, 8, 10, 15, and 20 years. Seven regional ethics review boards (Gothenburg, Lund, Linköping, Örebro, Karolinska Institute, Uppsala, Umeå) approved the study protocol and written or oral informed consent was obtained from all participants. This consent procedure was reviewed and approved by all seven review boards, reference number 184−90, date of decision June 6, 1990 (renewed approval, reference number T508-17, date of decision June 12, 2017). The SOS study's primary endpoint was overall mortality; secondary endpoints included diabetes and cardiovascular disease, while cancer incidence was not predefined. Approval to access cancer registry data was obtained later through a separate application (reference number S604-01, date of decision June 16, 2002). The SOS study is registered in ClinicalTrials.gov, NCT01479452.

The analyses presented here were conceived post hoc and were not specified in the original trial protocol or analysis plan (S2 Appendix). While the SOS study was designed as a prospective, controlled intervention trial, the present analyses were developed after trial completion of the analyses on primary outcome, to address additional hypotheses regarding surgery-induced weight loss and cancer risk. These analyses were prespecified, and a statistical analysis plan was prepared before undertaking them (S3 Appendix).

## Data collection

Baseline and follow-up data were collected through clinical examinations, questionnaires, and blood tests. Alcohol intake was calculated from dietary questionnaires. Postmenopausal status was defined by a "no" to "Do you still menstruate?", reported surgical menopause, or age >51 years if data were missing. Smoking was defined as answering "yes" to "Do you smoke daily?". Cancer, death, and emigration data were obtained by linking SOS participants to national registers. Information on deaths and emigration was obtained from the Swedish Population and Address Register. The Swedish Cancer Registry covers over 95% of malignant tumors, with 99% morphologically verified [18]. The Swedish Cause of Death Register is a high-quality essentially complete register of all official (underlying) cause of deaths in Sweden since 1952 [19]. Causes of death were also independently reviewed by two authors using case sheets and autopsy reports. If the cause of death determined in the study diverged from the official record, the study-determined direct cause was used, because the official cause of death in some cases reflects the underlying preventable cause (e.g., obesity) in the chain of events leading to death. However, a majority (>97%) of study-defined cancer-related deaths were also officially attributed to cancer in the Swedish Cause of Death register. Obesity-related cancers were defined according to the IARC [4], and include esophageal adenocarcinoma, postmenopausal breast cancer, cancers of the kidney, colon/rectum, gastric cardia, liver, gallbladder, pancreas, ovary and thyroid, endometrial cancer, multiple myeloma, and meningioma. Other cancers were classified as non-obesity-related. Female-specific cancers include breast, ovarian, endometrial, cervix, and all other gynecological cancers [14]. The cut-off date for the current report was December 31, 2022.

## Statistical analysis

BMI changes were analyzed with multilevel linear mixed-effects regression models assuming normally distributed errors, separately for men and women. The observations were considered nested within the individuals, and the confidence intervals (CIs) were calculated taking the repeated measurements into account. Treatment, time, and their interaction as well as age were considered fixed, and the individual patients were considered random effects. Due to the nature of the available data, missingness was expected to be negligible for the primary outcomes and exposures. Where data were missing, we assumed that values were missing at random, and all available observed data were included in the analyses. Patients were analyzed according to their received intervention (i.e., participants were included in their original study group until any bariatric surgery was performed in the control group or there was a change in, or removal of, the bariatric surgical procedure in the surgery group).

Participants were followed until the first occurrence of cancer, death, emigration, or end of register-based follow-up (December 31st, 2022), whichever came first. Participants who emigrated, died or remained event-free at the end of follow-up were treated as censored observations in the analyses. In addition, control-group participants who underwent bariatric surgery during follow-up and surgery-group patients who had a procedure restoring normal anatomy were also censored from the analyses at the time of these operations. Kaplan–Meier estimates were used to compare time to first cancer diagnosis or cancer-related death across treatment groups. Cox proportional hazards models were used to calculate hazard ratios (HRs) and 95% CIs for surgical treatment effects on overall, obesity-related, and female-specific cancers. The proportional-hazards assumption was evaluated by assessing the interaction between treatment and the logarithm of time. When the assumption was not met, Cox regression models were modified to include time-varying effects. In addition, a Restricted Mean Survival Time (RMST) approach was used to compare cancer-free time between the surgery and control groups.

To account for baseline differences between the surgery and control groups, analyses were adjusted for major cancer risk factors defined by the American Cancer Society; age, sagittal diameter (a proxy for intra-abdominal adiposity), alcohol consumption, and smoking, [20] as well as serum insulin levels, which have previously been associated with cancer risk in sub-analyses in the SOS study [14,15], as well as in the Longitudinal Assessment of Bariatric Surgery (LABS) study [13].

All covariates were selected a priori the actual analyses and included in the models regardless of their actual statistical significance.

To formally test whether baseline insulin modified the effect of bariatric surgery on cancer incidence and cancer-related mortality, we included an interaction term using the continuous insulin variable in Cox proportional hazards models. These analyses were also adjusted for other risk factors as described above. For descriptive purposes, cancer incidence and mortality were presented across data-driven tertiles of baseline insulin, selected to provide balanced subgroup sizes and to align with prior SOS analyses using quantile-based stratification. No established clinical insulin thresholds exists, however, the lowest insulin third [14,15] in the SOS cohort corresponds to approximately normal levels [21].

In sensitivity time-to-event analyses accounting for undiagnosed baseline cancer, follow-up time for all participants was initiated at year 3 after baseline, and only cancer events occurring after this initial 3-year follow-up period were included. In addition, participants with a history of cancer before study inclusion were excluded.

Analyses were performed using Stata version 18.0 (StataCorp, 2023. Stata Statistical Software: Release 18. College Station, TX: StataCorp LLC). RMSTs were analyzed with the Stata strmst2-package [22].

This study is reported as per the Strengthening the Reporting of Observational Studies in Epidemiology (STROBE) guideline (S1 STROBE Checklist).

**AI-assisted technologies:** During the preparation of this work the authors used Microsoft Copilot in order to correct the language and enhance readability in some sentences. After using this tool, the authors reviewed and edited the content as needed and take full responsibility for the content of the publication.

## Results

### Participants and weight changes during follow-up

The numbers of patients in the as-treated surgery and control groups were 2,007 and 2,040, respectively. Baseline characteristics of the patients, categorized by sex and treatment group, are shown in Table 1. The majority of the participants in both groups were women (71%). Although the two study groups were generally well balanced with respect to baseline

**Table 1. Baseline characteristics.**

|  | Women | | Men | |
|---|---|---|---|---|
|  | Surgery[1] N = 1,420 | Control N = 1,447 | Surgery[1] N = 587 | Control N = 593 |
| Age, years | 47.2 (6.0) | 48.7 (6.3) | 47.3 (5.8) | 48.6 (6.1) |
| Body mass index, kg/m² | 42.8 (4.3) | 40.7 (4.6) | 41.3 (4.8) | 38.6 (4.7) |
| Sagittal diameter, cm | 28.1 (3.4) | 26.9 (3.4) | 30.7 (3.7) | 28.6 (3.9) |
| Blood glucose, mmol/L | 5.1 (1.9) | 4.8 (1.7) | 5.5 (2.2) | 5.2 (2.1) |
| Insulin, mU/L | 19.9 (12.9) | 16.4 (9.6) | 25.4 (14.9) | 22.0 (14.2) |
| HbA1c, mmol/mol | 41.3 (12.0) | 39.8 (10.2) | 44.9 (13.8) | 42.6 (12.9) |
| T2D, N (%) | 203 (14.4) | 158 (10.9) | 141 (24.1) | 105 (17.7) |
| Alcohol intake, g/day | 3.2 (4.5) | 3.3 (5.1) | 10.0 (9.8) | 10.1 (11.3) |
| Daily smokers, N (%) | 366 (25.8) | 284 (19.7) | 152 (25.9) | 138 (23.5) |
| Previous cancer, N (%) | 105 (7.4) | 85 (5.9) | 3 (0.5) | 3 (0.5) |
| Year of inclusion | 1994.2 (3.3) | 1994.5 (3.5) | 1993.9 (3.4) | 1993.9 (3.4) |
| Postmenopausal, N (%) | 433 (30.5) | 529 (36.6) |  |  |

Values are presented as mean (standard deviation) unless otherwise indicated.

[1]Women/men in the surgery group underwent non-adjustable or adjustable gastric banding (N = 260/116), vertical banded gastroplasty (N = 970/395), or gastric bypass (N = 190/76). T2D, Type 2 diabetes; HbA1c, Hemoglobin A1c.

characteristics, measures of adiposity and several cardiovascular risk factors were less favorable in the surgery group (Table 1). On average, the surgery group was younger, had higher BMI, larger abdominal sagittal diameter, higher glucose, HbA1c and insulin levels, and higher prevalence of diabetes compared to the control group. These differences are primarily attributable to weight changes that occurred during the interval between matching and baseline measurements: while awaiting bariatric surgery (on average more than one year), participants in the surgery group tended to gain weight, whereas control subjects tended to lose weight [23]. Women in the surgery group were more often smokers (25.8% versus 19.7%) and had slightly higher prior cancer rates (7.4% versus 5.9%) compared to female control participants. Baseline alcohol intake did not differ between groups (Table 1). Average weight changes during 20 years of follow-up, stratified by sex and treatment group, are shown in Fig B in S1 Appendix.

On the date of analysis, the follow-up time was up to 35.1 years, with a median follow-up of 26.8 (interquartile range (IQR) 22.9–29.6) years in the surgery group and 24.9 (IQR 18.7–28.8) years in the control group.

## Overall cancer incidence in women and men

In women, there were 355 first-time cancers in the surgery group and 407 in the control group (Fig 1A, left panel). The cancer incidence rate (IR) per 1,000 person-years was 11.0 (95% CI [9.9, 12.2]) in the surgery group and 14.0 (95% CI [12.7, 15.4]) in controls. Over 31.5 years of follow-up, the difference in restricted mean cancer-free time between the surgery and control groups was 1.4 years (95% CI [0.8, 2.1]; $p < 0.001$). The unadjusted HR with surgery was 0.75 (95% CI [0.65, 0.87]; $p < 0.001$), and after multivariable adjustments, the adjusted hazard ratio (HRadj) was 0.78 (95% CI [0.67, 0.90]; $p = 0.001$). The association between surgery and cancer incidence in women was time-dependent, being more pronounced early during follow-up. Sensitivity analyses produced similar estimates when follow-up was started at year 3 after baseline (HRadj = 0.82; 95% CI [0.70, 0.95]; $p = 0.010$) (Fig C in S1 Appendix), and after excluding n = 190 with prior cancer (HRadj = 0.82; 95% CI [0.70, 0.97]; $p = 0.017$) (Fig D in S1 Appendix).

In men, bariatric surgery was not associated with cancer incidence (HRadj = 1.16; 95% CI [0.92, 1.46]; $p = 0.214$), based on 165 events (IR = 13.4; 95% CI [11.5, 15.6]) in the surgery group and 145 events in the control group (IR = 12.2; 95% CI [10.3, 14.3]) (Fig 1A, right panel). The association between surgery and cancer incidence differed by sex (adjusted p for sex-treatment interaction = 0.013; accounting for non-proportional hazard $p = 0.011$).

## Cancer-related mortality in women and men

In women, there were 122 cancer-related deaths in the surgery group and 149 in the control group. Surgery was associated with a lower rate of mortality over time (HR = 0.73; 95% CI [0.57, 0.92]; $p = 0.009$), although the association was attenuated after adjustment (HRadj = 0.78; 95% CI [0.61, 1.00]; $p = 0.050$) (Fig 1B, left panel). In men, no association was observed (HR = 0.87; 95% CI [0.60–1.25]; $p = 0.439$) (Fig 1B, right panel), and there was no evidence of a sex-treatment interaction for mortality (adjusted $p$ for interaction = 0.433).

## Incidence and mortality rates of obesity-related and female-specific cancers in women

In women, 60% of incident cancers were obesity-related (Table A in S1 Appendix), with 214 cases in the surgery group and 276 in the control group (Fig 2A). Bariatric surgery was associated with lower rate of obesity-related cancer over time (HRadj = 0.70; 95% CI [0.58, 0.85]; $p < 0.001$). However, no association was observed for obesity-related cancer mortality (76 versus 85 deaths; HRadj = 0.82; 95% CI [0.59, 1.12]; $p = 0.211$) (Fig 2B), or for the incidence of non-obesity-related cancers (Fig E in S1 Appendix).

Given the high prevalence of female-specific cancers (Table A in S1 Appendix), we also examined the association between bariatric surgery and their incidence and mortality. During follow-up, 139 female-specific cancers occurred in the surgery group and 204 in the control group (Fig 2C), with a corresponding HRadj of

## A. Cancer incidence

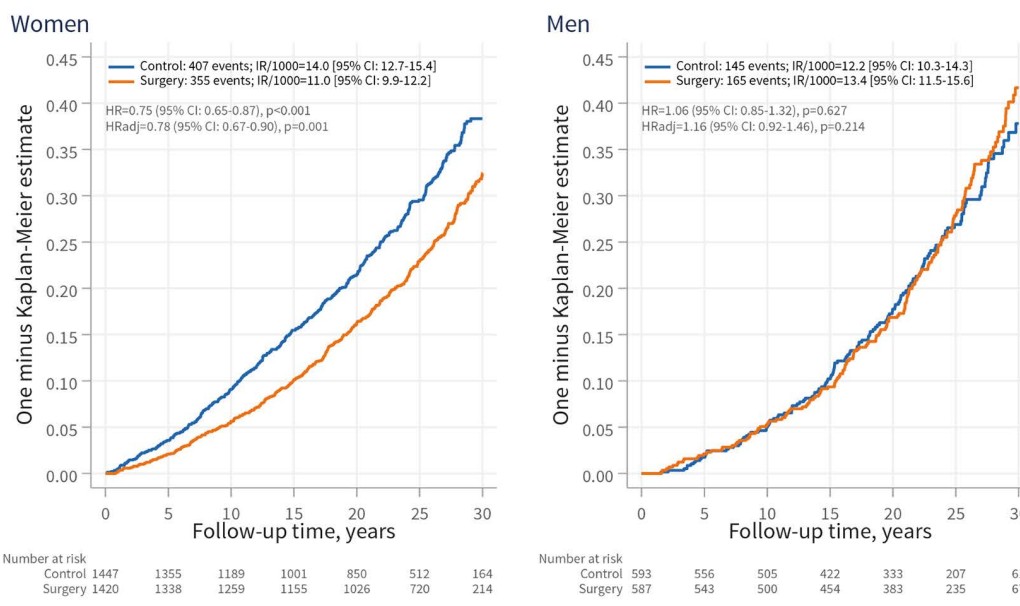

## B. Cancer-related mortality

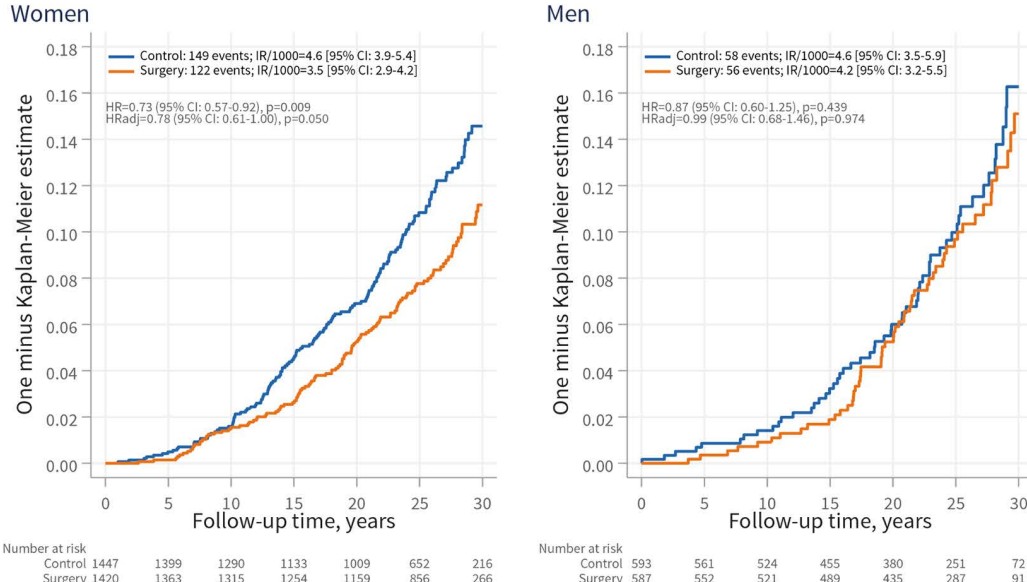

**Fig 1. Cumulative incidence of overall cancer (A) and cancer-related mortality (B) in control participants and patients undergoing surgery, stratified by sex.** IR/1,000, incidence rate per 1,000 person-years; 95% CI, 95% confidence interval: HR, hazard ratio; HRadj, HR adjusted for age, sagittal diameter, alcohol consumption, smoking, and serum insulin levels.

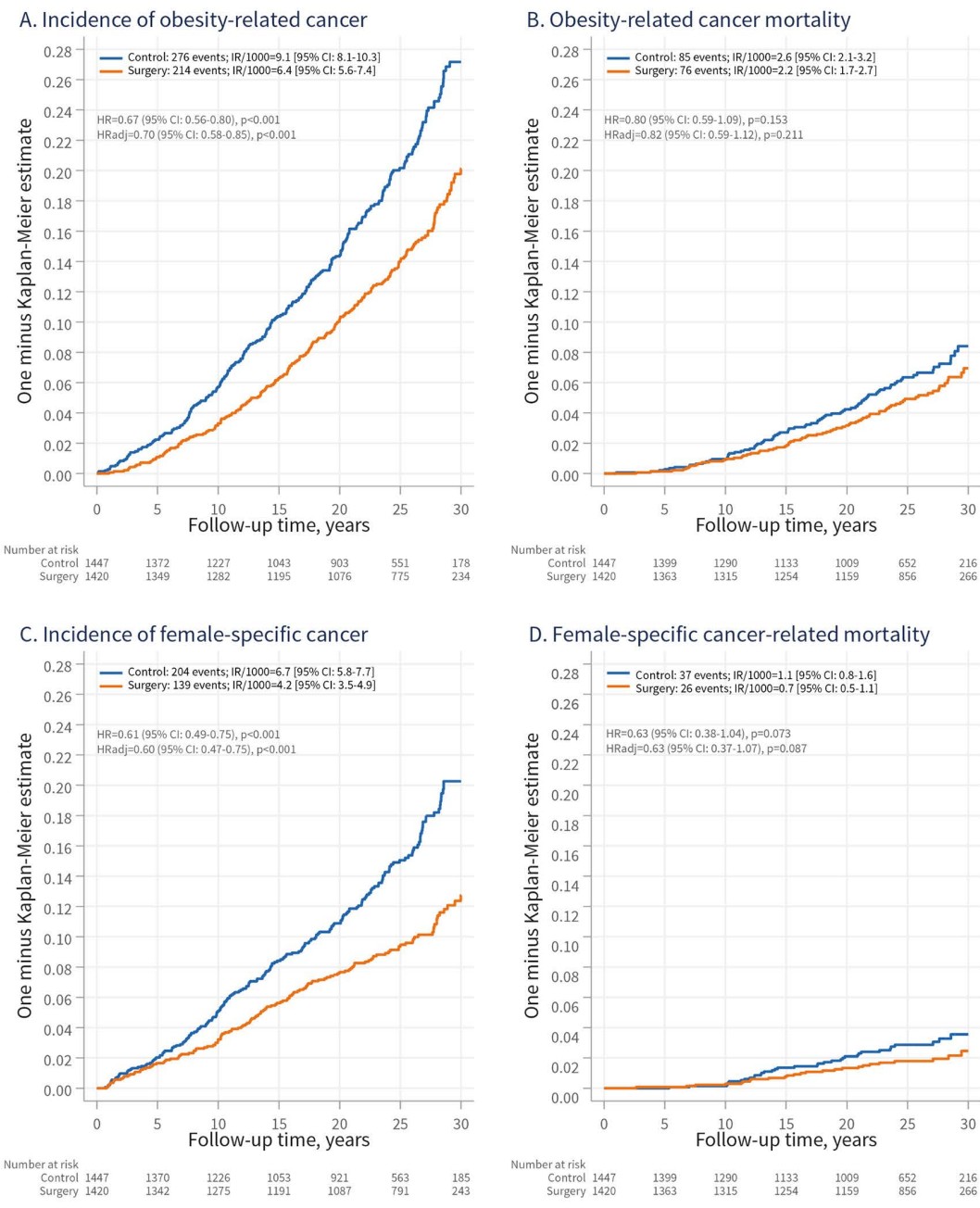

**Fig 2. Cumulative incidence of obesity-related and female-specific cancer (A, C) and cancer-related mortality (B, D) in women from the surgery and control groups.** IR/1,000, incidence rate per 1,000 person-years; 95% CI, 95% confidence interval: HR, hazard ratio; HRadj, HR adjusted for age, sagittal diameter, alcohol consumption, smoking, and serum insulin levels.

0.60 (95% CI [0.47, 0.75]; *p* < 0.001). For female-specific cancer mortality (26 versus 37 deaths), the HR with surgery was in a similar range, but the association was only suggestive (HRadj = 0.63; 95% CI [0.37, 1.07]; *p* = 0.087). We found no evidence that bariatric surgery was associated with non-female-specific cancer incidence (Fig E in S1 Appendix).

## Cancer incidence and cancer mortality in women stratified by baseline insulin levels

We previously reported that high baseline insulin levels were associated with a greater treatment benefit of bariatric surgery for female-specific cancers [14,15]. To explore this further, women were grouped by insulin tertiles (Fig 3). Among female controls, cancer IRs increased across insulin tertiles, ranging from 11.6 cases per 1,000-person-years (95% CI [9.9, 13.7]) in the lowest third to 17.8 cases per 1,000-person-years (95% CI [15.0, 21.2]) in the highest third (Fig 3A).

Next, we assessed the overall cancer IRs after bariatric surgery versus controls, across the insulin subgroups. Surgery was associated with a lower rate of overall cancer in the two highest thirds (HRadj = 0.72; 95% CI [0.55, 0.93]; $p$ = 0.011, and HRadj = 0.68; 95% CI [0.53, 0.86]; $p$ = 0.002, respectively), but not in the lowest third (HRadj = 0.99; 95% CI [0.75, 1.30]; $p$ = 0.919) (Fig 3A). A similar pattern emerged for obesity-related cancers: surgery was associated with lower rates in the two highest insulin thirds (HRadj = 0.52; 95% CI [0.37, 0.73]; $p$ < 0.001, and HRadj = 0.66; 95% CI [0.49, 0.88]; $p$ = 0.006) (Fig 3B). Notably, bariatric surgery was also associated with a lower rate of non-obesity-related cancers in the subgroup with the highest insulin levels (HRadj = 0.68; 95% CI [0.47, 0.99]; $p$ = 0.045) (Fig F in S1 Appendix). Treatment-insulin interactions were suggestive for overall and obesity-related cancers (adjusted interaction $p$ = 0.055 and 0.072, respectively), but no evidence of interaction was observed for non-obesity-related cancers (adjusted interaction $p$ = 0.469).

We then examined IRs of female-specific cancers across insulin subgroups. Surgery was associated with a lower rate of female-specific cancers in the two highest thirds (HRadj = 0.48; 95% CI [0.33, 0.72]; $p$ < 0.001, and HRadj = 0.49; 95% CI [0.34, 0.70]; $p$ < 0.001) (Fig 3C) with evidence of heterogeneity by insulin level (adjusted treatment–insulin interaction $p$ = 0.021). For non-female-specific cancers, no association between surgery and cancer incidence was observed in either of the insulin subgroups (Fig F in S1 Appendix).

We next examined cancer mortality across insulin subgroups in women. Compared with controls, surgery was associated with a lower rate of overall cancer mortality in the highest insulin third (HRadj = 0.64; 95% CI [0.43, 0.97]; $p$ = 0.035), but not in the lower two (Fig 4A), and there was no evidence of a treatment-insulin interaction ($p$ = 0.278). No association was observed between surgery and obesity-related cancer mortality across insulin subgroups (Fig 4B). In contrast, surgery was associated with a substantially lower rate of mortality from female-specific cancers in the highest insulin third (HRadj = 0.37; 95% CI [0.16, 0.85]; $p$ = 0.019), with evidence of heterogeneity by insulin level (adjusted treatment–insulin interaction $p$ = 0.039) (Fig 4C).

## Cancer incidence in women after bariatric surgery, stratified by one-year weight loss

Retrospective studies suggest that cancer risk may relate to degree of weight loss [7,16]. To test this, women in the surgery group were stratified by 1-year weight loss tertiles. No association was found between weight loss and incidence of overall, obesity-related, or female-specific cancer (Fig G in S1 Appendix).

## Discussion

Our updated analysis of long-term cancer outcomes in the prospective controlled SOS study confirm that surgery-induced weight loss is associated with a lower IR of cancer in women, whereas no association is observed in men. Furthermore, the strongest association was seen for female-specific cancers among women with high baseline insulin levels, highlighting insulin status as a potential modifier of long-term cancer risk.

It is now well established that surgery-induced weight loss is associated with a markedly reduced risk of cancer and cancer mortality [6,7,9,10,12,24]. The SOS study previously reported that surgery-induced weight loss appeared to be associated with reduced cancer risk, but only in women [10]. After extending the follow-up period by another 17 years, surgery was associated with a 22% lower hazard (HRadj = 0.78) of incident cancer among women, with a strong indication of sex differences in treatment outcomes, aligning with associations reported in earlier retrospective studies [6,9,12]. Our results also show that surgery was associated with a 22% lower hazard (HRadj = 0.78) of cancer mortality among

### A. Overall cancer

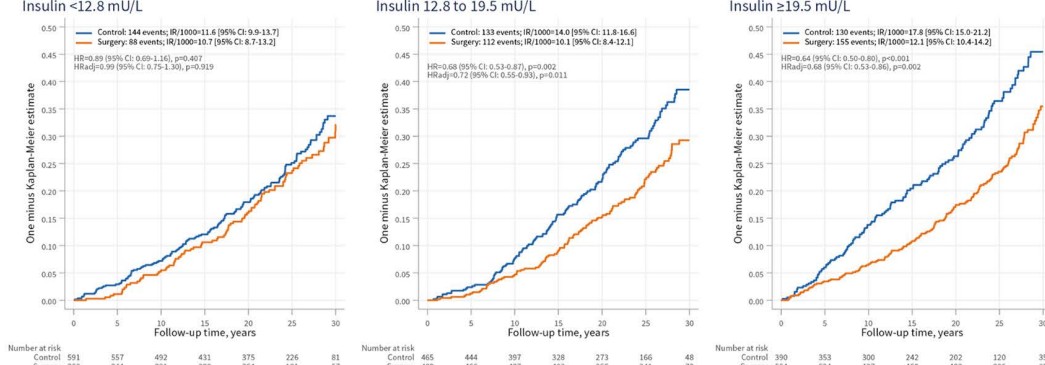

### B. Obesity-related cancer

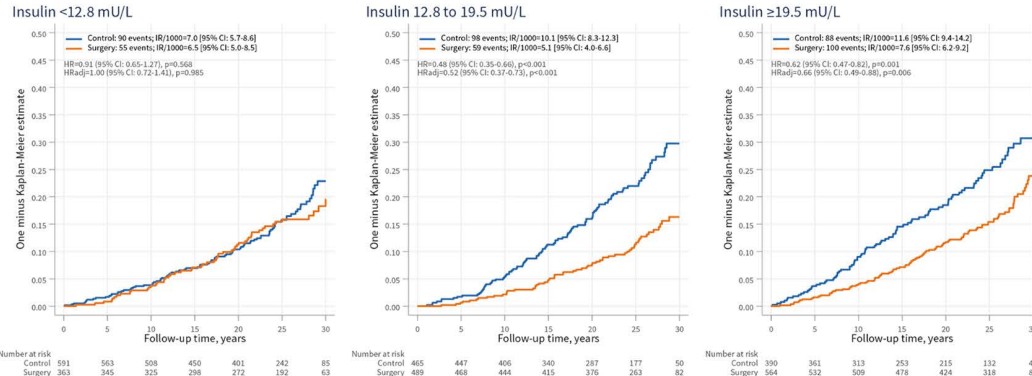

### C. Female-specific cancer

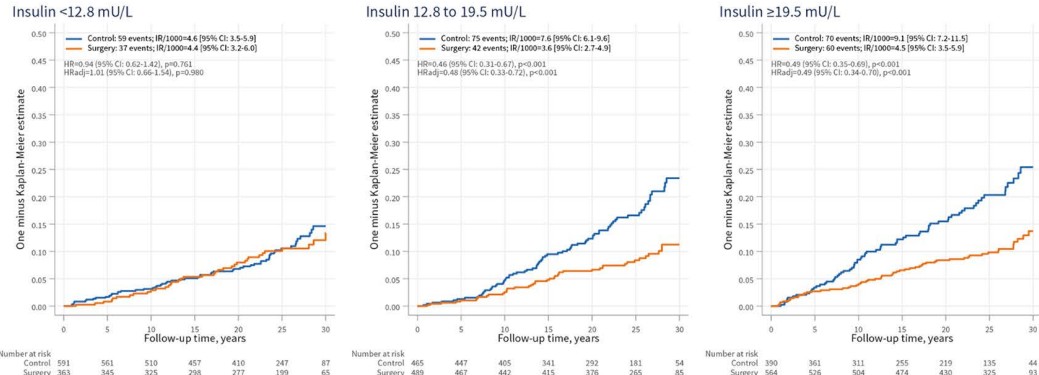

**Fig 3. Cumulative incidence of overall (A), obesity-related (B), and female-specific (C) cancer in women from the surgery and control groups, stratified by baseline insulin levels.** IR/1,000, incidence rate per 1,000 person-years; 95% CI, 95% confidence interval: HR, hazard ratio; HRadj, HR adjusted for age, sagittal diameter, alcohol consumption, smoking, and serum insulin levels.

### A. Overall cancer-related mortality

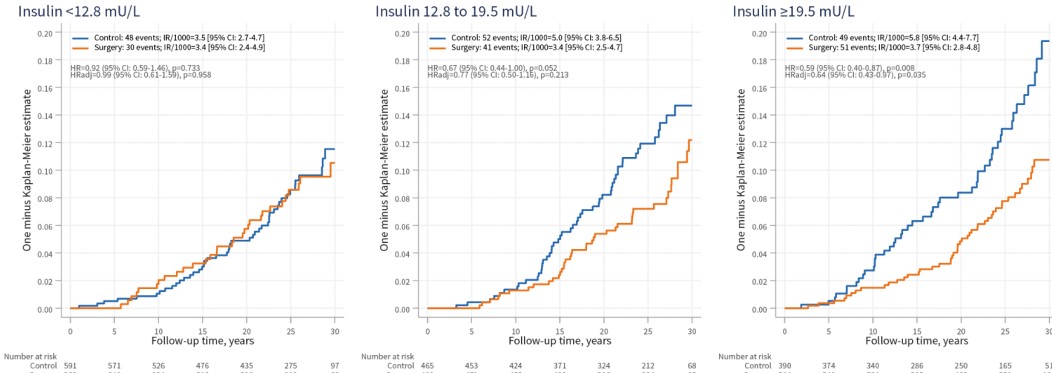

### B. Obesity-related cancer mortality

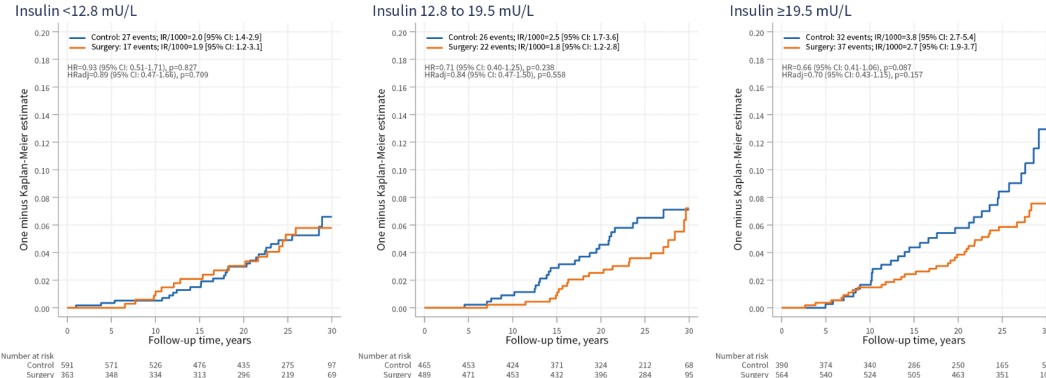

### C. Female-specific cancer-related mortality

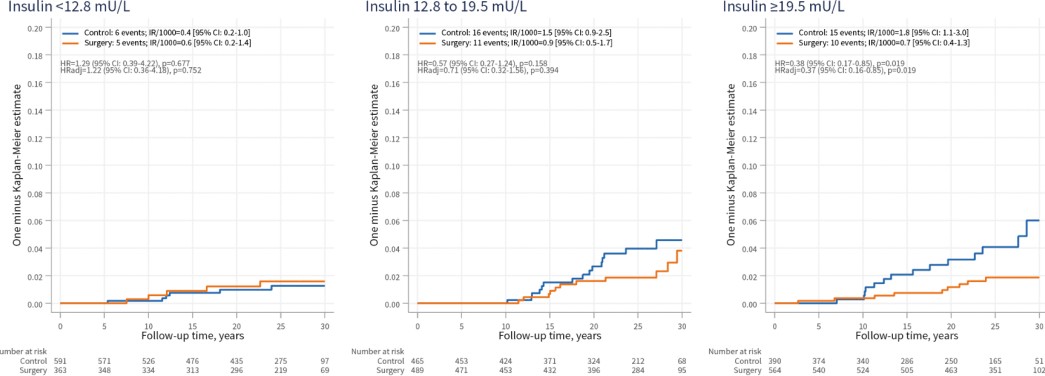

**Fig 4. Overall (A), obesity-related (B), and female-specific (C) cancer-related mortality in women from the surgery and control groups, stratified by baseline insulin levels.** IR/1,000, incidence rate per 1,000 person-years; 95% CI, 95% confidence interval; HR, hazard ratio; HRadj, HR adjusted for age, sagittal diameter, alcohol consumption, smoking, and serum insulin levels.

women, while no clear association was observed in men, consistent with retrospective findings from the Utah Population Database [6].

At present, there is evidence supporting a link between obesity and an increased risk of 13 cancers, including common female cancers such as breast and endometrial cancer [4]. In this report, surgery was associated with a 30% lower hazard (HRadj = 0.70) of obesity-related cancers in women, with no association seen for non-obesity-related cancers. The definition of obesity-related cancers may warrant discussion, as additional cancers may be included over time. It should be noted that the IARC report listed cancers like diffuse large B cell lymphoma as potentially obesity-related but found the evidence insufficient at the time [4]. In the SOS study, we previously showed that surgery-induced weight loss was associated with lower incidence of malignant melanoma and lymphomas, both currently classified as non-obesity-related cancers [25,26]. Moreover, a recent population-based study suggested adding several rare or subtype-specific cancers to the obesity-related list [27]. However, the rarity of these cancers makes it difficult to detect clear associations with obesity or the effects of weight loss.

Cancer risk in individuals with obesity is multifactorial and influenced by a complex interplay between hormones, the immune system, chronic inflammation, bile acids, and the gut microbiome, which together modulate key signaling pathways [28]. Notably, findings from the prospective, though uncontrolled, LABS study suggest that reductions in insulin, leptin, and ghrelin after bariatric surgery are independently associated with decreased cancer risk [13]. These hormones have all been implicated in tumor biology and may contribute to cancer progression [29–31]. Of note, they also exhibit sex differences in the context of obesity [32,33], reflecting distinct metabolic profiles that could underlie sex-specific associations between weight loss and cancer risk. In addition, body composition and fat distribution are sexually dimorphic. Men typically accumulate more visceral adipose tissue, whereas women predominantly store subcutaneous fat. These sex-specific patterns of fat distribution are associated with differences in insulin sensitivity, systemic inflammation, and potentially cancer susceptibility [33]. Furthermore, many female-specific cancers, such as breast and endometrial cancer, are estrogen-dependent. Of particular relevance, insulin and estrogen interact in ways that may promote cancer development and progression, including reciprocal receptor activation leading to an increased fraction of bioactive estrogen [28].

The global obesity epidemic not only increases overall cancer prevalence but may also disproportionately increase women's susceptibility to cancer-related morbidity and mortality. On a global scale, the most prevalent obesity-related cancers, endometrial, postmenopausal breast, and colon cancers, constitute about two-thirds of the cancers associated with excess BMI [28]. Women with obesity are not only more likely to develop cancer compared to women with lower BMI, but are also less likely to undergo screening, often diagnosed at a later stage, and have worse cancer survival [34]. Understanding how obesity affects cancer risk in women, and how weight loss mitigates that risk, therefore holds significant public health importance. Previous analyses of women in the SOS study suggested that hyperinsulinemia at the time of surgery is associated with a greater reduction of cancer risk [14,15]. We now extend these findings by demonstrating that ascending baseline insulin levels were associated with a gradual increase in cancer incidence in the control group, and that surgery was associated with 32% lower hazard (HRadj = 0.68) of overall cancer and a 34% lower hazard (HRadj = 0.66) of obesity-related cancer among women with high baseline insulin. Notably, the reduction in the rate of non-obesity-related cancers over time was of a similar magnitude, with surgery associated with a 32% lower hazard (HRadj = 0.68) in the highest insulin third. In addition, for female-specific cancers, the association was even more pronounced in the highest third (51% lower hazard; HRadj = 0.49). Conversely, no association with surgery was observed for non-female-specific cancers, regardless of baseline insulin level. Taken together, these findings suggest that tailoring weight loss interventions for women with obesity and hyperinsulinemia could enhance cancer prevention strategies.

Retrospective and uncontrolled studies have proposed that the degree of weight loss after bariatric surgery may influence cancer risk reduction [7,16,35], with one study indicating a weight loss threshold [13] for measurable benefits post-surgery. In contrast, our analysis detected no differences in cancer incidence across 1-year weight loss thirds in patients who underwent surgery, consistent with our previous findings [10]. However, our study might have too limited

range of weight loss within the surgery group and therefore be underpowered to detect an association. Furthermore, the analyses could have been influenced by the low proportion of SOS patients who underwent gastric bypass, a procedure that may induce more pronounced changes in cancer-related metabolic and hormonal risk factors compared with older surgical methods [7,35]. Additionally, other differences in study design further complicate comparisons; retrospective studies using registry data may be affected by unmeasured confounders [36], and studies with shorter follow-up or smaller sample sizes may fail to capture long-term risk reductions. Genetic predisposition may also affect weight loss outcomes and cancer risk, with some individuals experiencing greater benefits, as shown by us and others [37,38]. New obesity drugs, originally developed for diabetes treatment, now rival bariatric surgery in efficacy [39], but evidence regarding possible associations with cancer risk remains limited [11]. To this end, long-term data on cancer risk following surgery-induced weight loss, with follow-up far exceeding that of obesity drug trials, could provide valuable insights. Such evidence may, for example, inform future strategies for integrating emerging weight-loss therapies into precision medicine approaches for obesity management and cancer prevention [40].

Major strengths of the SOS study include its long follow-up, prospective design, and access to detailed national register data. Considering the extensive cohort size and long follow-up period required to evaluate cancer outcomes, the SOS study remains a critical source of knowledge in this area. Importantly, all participants, regardless of treatment choice, were eligible for surgery, a design shown to minimize unmeasured confounding and selection bias in observational studies [41]. However, there are also limitations. First, when the SOS study was initiated, obesity had not yet been linked to increased cancer risk. Consequently, cancer incidence was not a predefined endpoint, and all related analyses are exploratory. Second, despite the comprehensive matching strategy, residual confounding cannot be entirely excluded. However, because the surgery group had more adverse baseline risk factors than the controls, any bias introduced by these differences would likely have led to an underestimation, rather than an overestimation, of the benefits of surgery. Third, because the SOS study was initiated in the 1980s, most participants underwent surgical procedures that are no longer commonly performed. As a result, the number of patients treated with gastric bypass was limited, and none underwent sleeve gastrectomy. This restricts our ability to assess modern procedure-specific effects on cancer risk. Moreover, the potential cancer-preventive effects of newer obesity treatments, including glucagon-like peptide-1 receptor agonists, remains to be determined. Thus, long-term studies involving contemporary surgical techniques and pharmacotherapies are needed to evaluate the generalizability of our findings. Finally, the number of men in the SOS study is considerably lower than the number of women, affecting the sex-specific analyses.

In conclusion, our findings underscore that surgery-induced weight loss is associated with a lower risk of cancer and cancer-related mortality in women with obesity, with the strongest associations observed for female-specific cancers and mortality caused by these cancers in women with high insulin levels. These results also point to the potential value of integrating sex and metabolic profiling into targeted weight-loss strategies for cancer prevention.

## Supporting information

**S1 STROBE Checklist. STrengthening the Reporting of OBservational studies in Epidemiology (STROBE) Statement—checklist of items that should be included in reports of observational studies, available at** https://www.strobe-statement.org/**, licenced under CC BY 4.0.**
(DOCX)

**S1 Appendix. Table A.** Obesity-related and non-obesity-related cancer events by sex and treatment. **Fig A.** Flow diagram of patient recruitment in the SOS study. **Fig B.** Changes in body mass index over 20 years in control participants and patients undergoing surgery, stratified by sex. **Fig C.** Sensitivity analysis of overall cancer incidence in women with follow-up starting 3 years after baseline. **Fig D.** Sensitivity analysis of overall cancer incidence in women with follow-up starting 3 years after baseline; individuals with cancer events prior to baseline excluded. **Fig E.** Incidence of

non-obesity-related and non-female-specific cancer incidence in women from the surgery and control groups. **Fig F.** Incidence of non-obesity-related, and non-female-specific cancer in women from the surgery and control groups, stratified by baseline insulin levels. **Fig G**. Incidence of overall, obesity-related, and female-specific cancer from start of year 4 and onwards, in women from the surgery group, stratified by 1-year weight loss.
(DOCX)

**S2 Appendix. Original SOS study protocol.** The SOS study was designed as a prospective, controlled intervention trial. The present analyses (insulin and cancer outcomes) were post hoc and were not prespecified in the original protocol.
(PDF)

**S3 Appendix. Statistical analysis plan.**
(PDF)

## Acknowledgments

We thank the staff members at the 480 primary healthcare centers and 25 surgical departments in Sweden that participated in the Swedish Obese Subjects (SOS) study.

## Author contributions

**Conceptualization:** Kajsa Sjöholm, Markku Peltonen, Lena M. S. Carlsson, Magdalena Taube.

**Data curation:** Peter Jacobson, Markku Peltonen, Magdalena Taube.

**Formal analysis:** Markku Peltonen.

**Funding acquisition:** Kajsa Sjöholm, Per-Arne Svensson, Sofie Ahlin, Lena M. S. Carlsson, Magdalena Taube.

**Investigation:** Kajsa Sjöholm, Per-Arne Svensson, Johanna C. Andersson-Assarsson, Peter Jacobson, Sofie Ahlin, Cecilia Karlsson, Björn Carlsson, Felipe M. Kristensson, Per Karlsson, Markku Peltonen, Lena M. S. Carlsson, Magdalena Taube.

**Methodology:** Kajsa Sjöholm, Markku Peltonen.

**Project administration:** Kajsa Sjöholm, Magdalena Taube.

**Resources:** Kajsa Sjöholm, Per-Arne Svensson, Johanna C. Andersson-Assarsson, Lena M. S. Carlsson, Magdalena Taube.

**Software:** Markku Peltonen.

**Supervision:** Kajsa Sjöholm, Markku Peltonen, Lena M. S. Carlsson, Magdalena Taube.

**Visualization:** Markku Peltonen.

**Writing – original draft:** Kajsa Sjöholm, Magdalena Taube.

**Writing – review & editing:** Kajsa Sjöholm, Per-Arne Svensson, Johanna C. Andersson-Assarsson, Peter Jacobson, Sofie Ahlin, Cecilia Karlsson, Björn Carlsson, Felipe M. Kristensson, Per Karlsson, Markku Peltonen, Lena M. S. Carlsson, Magdalena Taube.

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
