## [Editor Report · Decision Letter 0]

30 Jun 2025

Dear Dr Taube,

Thank you for submitting your manuscript entitled "Sex-specific association between baseline insulin levels and cancer outcomes after surgery-induced weight loss: The prospective, controlled Swedish Obese Subjects study" for consideration by PLOS Medicine.

Your manuscript has now been evaluated by the PLOS Medicine editorial staff and I am writing to let you know that we would like to send your submission out for external peer review.

For clinical studies, please upload a copy of your trial study protocol as a supporting information file. The study protocol should be the version submitted for approval to the institutional review board or ethics committee, should include any amendments to the study protocol, as well as the date of their approval by the institutional review or ethics committee. Please also detail any deviations from the study protocol in the Methods section of your manuscript. The editors will consider the protocol and study conduct prior to a final decision for external review.

Please re-submit your manuscript within two working days, i.e. by Jul 02 2025.

Feel free to email me at atosun@plos.org or us at plosmedicine@plos.org if you have any queries relating to your submission.

Kind regards,

Alexandra Tosun, PhD

Senior Editor

PLOS Medicine

---

## [Decision Letter · Decision Letter 1]

25 Aug 2025

Dear Dr Taube,

Many thanks for submitting your manuscript "Sex-specific association between baseline insulin levels and cancer outcomes after surgery-induced weight loss: The prospective, controlled Swedish Obese Subjects study" (PMEDICINE-D-25-02306R1) to PLOS Medicine. The paper has been reviewed by subject experts and a statistician; their comments are included below and can also be accessed here: ********

As you will see, the reviewers responded positively to the study and provided valuable feedback to strengthen the manuscript further. After discussing the paper with the editorial team and an academic editor with relevant expertise, I'm pleased to invite you to revise the paper in response to the reviewers' comments. We plan to send the revised paper to some or all of the original reviewers, and we cannot provide any guarantees at this stage regarding publication.

We ask that you submit your revision by Sep 15 2025. However, if this deadline is not feasible, please contact me by email, and we can discuss a suitable alternative.

Don't hesitate to contact me directly with any questions (atosun@plos.org).

Best regards,

Alexandra

Alexandra Tosun, PhD

Senior Editor

PLOS Medicine

atosun@plos.org

Comments from the editorial team:

We recommend avoiding the term "female-specific cancers" and using "gynecologic cancers" instead.

Comments from the reviewers:

Reviewer #1:

This is a well-conducted and timely study leveraging the long-term data from SOS) cohort to explore sex-specific cancer outcomes after bariatric surgery, with a focus on baseline insulin levels. The manuscript is clearly written, the analyses are rigorous, and the findings are both clinically and biologically plausible. The potential for these results to inform risk-stratified preventive oncology strategies is high. However, somel concerns related to interpretation, framing, and generalizability merit attention.

MAJOR POINTS

1. Clarity on Causality and Stratification: The manuscript occasionally overstates causality or generalizability given the non-randomized design, please clarify throughout the manuscript that the findings reflect associations, not causal inferences. Consider replacing phrases like "surgery reduced cancer risk" with "was associated with a lower cancer risk."

2. Sex Differences Interpretation; despite the emphasis on sex-surgery interactions,the biological basis remains underexplored. Expand discussion on plausible biological reasons for sex-specific effects ( hormonal, adipose distribution, cancer types), or acknowledge this as an open question.

3. Insulin as a Modifier: insulin is proposed as a predictor of benefit, only female-specific cancers had significant treatment-insulin interaction p-values. Soften the conclusion that insulin is a "predictor of benefit" across all cancer types. Emphasize where the interaction was statistically robust and where it was only suggestive.

4. Control Group Match: The matching strategy is not clearly explained in the manuscript body. Include a concise explanation of the matching variables and discuss residual confounding, especially given significant baseline differences.

5. Weight Loss Degree Analysis: Acknowledge the null finding regarding 1 year WL limitation more explicitly. Discuss whether type of surgery (which affects weight loss and hormonal response) may confound this analysis.

6. Generalizability: most SOS surgical procedures are obsolete. It is worth a concise discussion on extrapolating findings to modern procedures (RYGB, SG) or GLP-1-based pharmacotherapy.

MINOR ISSUES: Consider adding risk tables below Kaplan-Meier curves for clarity. Define "female-specific cancers" in the Abstract and Key Points. Emphasize again in the Discussion that cancer was not a predefined endpoint

Reviewer #2: The Swedish Obese Subjects (SOS) prospective cohort study recruited weight loss surgery patients and matched controls, and previously published analyses of this cohort found reduced cancer incidence in women but not in men. This study presents similar analyses based on the same cohort with 17 additional years of follow up and considering additional predictors. The study confirmed earlier findings of reduced cancer incidence and mortality in women but not in men, with the effect further mediated by baseline insulin level.

Overall, this manuscript is an excellent initial submission. It is clearly written, logically presented, and cogently argued. That said, there are some areas of methodological concerns and opportunities to improve clarity. I'd also encourage the authors to complete and include a STROBE checklist (https://www.equator-network.org/reporting-guidelines/strobe/) with their next submission.

Detailed feedback on the manuscript is provided below. My primary focus has been methodological, but I have made a few broader points as well. All items are major unless indicated otherwise.

1. The Introduction is both comprehensive and succinct. Very well done.

2. [MINOR] Consider presenting the proportion of causes of death where the official and study causes differed.

3. Generally, p-values should not be used in descriptive tables (see section 14a of the expanded STROBE guidelines https://journals.plos.org/plosmedicine/article?id=10.1371/journal.pmed.0040297). Suggest the authors remove from Methods and Table 1.

4. I was pleased to see the authors considering missingness in their methods but did want to flag that missing at random (MAR) is a relatively strong and untestable assumption. I suggest the authors provide further justification, including any predictors of missingness the authors considered and whether they were included in the models.

5. The descriptions of the statistical modelling approaches in the Methods are a good start, but more detail is needed. If modelling results are presented, then enough detail should be included in the Methods so an interested reader can replicate the analyses. Specifically, while the authors have noted the model type they have used (e.g., "multilevel mixed-effect (linear) regression" and Cox PH), it's important to clearly state the variables in the models (fixed and random effects for hierarchical models, cluster variables for GEE etc.). Including abridged modelling code as a supplement can also help when more complex analyses are being undertaken.

6. Relatedly, I found the manuscript a little muddled when it came to stratified analyses vs. analyses controlling for variables (with or without interactions). Suggest the authors make this clearer in their Methods and Results.

7. Consider explaining how variables included in the statistical models were chosen e.g., prior research, expert knowledge, DAG.

8. I am not at all clear how case-fatality rates were calculated and, in the Results, why no statistics are presented. Suggest the authors clarify methods and present analytical results (or justify their absence).

9. [MINOR] I found the sentence in the Statistical Analysis section starting "Cancer incidence and cancer mortality were tracked…" a little hard to read. Consider rewording as being clear on events and censoring criteria is crucial.

10. While hazard ratios are the generally accepted statistic for time-to-event analyses, they have some well-recognised limitations (see https://doi.org/10.1093/ehjacc/zuae017 and https://evidence.nejm.org/doi/10.1056/EVIDe2300142). The authors may wish to consider supplementing key HR outcomes with another statistic like difference in median survival or restricted mean survival time to assist with interpretability.

11. Given the long follow-up, I suggest the authors check that the proportional hazard (PH) assumption remains valid when using Cox regression (e.g., Schoenfeld residuals). If PH is violated, consider adjusting the Cox model (e.g., including time-varying covariates) or using alternative approaches that do not assume proportional hazards (e.g., RMST).

12. It's not clear to me how the authors handled patients with a history of cancer before inclusion into the study nor patients who experienced a cancer event in the first three years. I see from the methods the authors conducted sensitivity analyses excluding "… cancer events within the first three years and pre-inclusion events." Does this mean in the main analyses patients with a history of cancer are contributing time as if they were cancer free? In the sensitivity analyses was the event dropped but the patient kept, or was the patient dropped entirely. I suggest the authors make their methods clearer here and carefully consider any risk of bias arising from their choices.

13. When presenting results from a Cox regression (e.g., Fig 1), if a p-value is to be presented in addition the to the HR and CI then the p-value should come from the Cox model, not a log-rank test on the same data. The log-rank test can be used in a simpler, unadjusted analyses but mixing a log-rank test p-value with Cox regression results is confusing at best and misleading at worst.

14. The authors present many p-values and make multiple references to "(statistical) significance". This is risky because the more claims of significance are made, the greater the risk of false positives and the more important it becomes to consider multiple testing correction strategies like the Bonferroni adjustment. Assuming the authors would prefer to avoid this, I'd suggest excluding p-values, instead just presenting point estimates and 95% CIs. I'd also suggest moderating or changing language away from statistical significance as an all-or-nothing threshold. One option is to talk in terms of strength of evidence e.g., "strong/moderate/some evidence of…" or "no evidence was found of…".

15. While historically categorising continuous variables has been common practice, recent methodological research discourages this ( https://www.bmj.com/content/332/7549/1080.1 ). Suggest the authors consider analysing continuous variables as-is, including insulin and 1-year weight loss.

16. "After extending the follow-up period by another 17 years, we now observe a 22% reduction in post-surgery cancer risk among women…" This is not correct, hazard ratios and risk/probabilities are not interchangeable because hazard ratios reflect relative rates over time rather than absolute differences in risk. Suggest rewording here and anywhere else hazard ratios are interpreted as relative risk.

17. I think there are two important limitations to this study that were not identified by the authors. Patient characteristics like smoking and alcohol consumption are important cancer risk factors that may be susceptible to self-report bias. Given the larger proportion of women in the study, the study may be underpowered to detect treatment effects in men. Suggest the authors consider these for inclusion.

18. Aside from the few items above, the Discussion is well-constructed, clearly follows from the presented results, and is convincingly argued. Great work!

Reviewer #3: Dear Dr. Taube and colleagues,

This is an interesting work, studying the very long term medical outcomes of bariatric surgery on cancer. I find the results surprising and interesting, in particular the sex dependent effects of the surgery and effects on female cancers. The effects size are meaningful, the results statistically signifnicant, and I believe they will be of interest to the bariatric and surgical community. As for the cancer community, it is hard to say, there are many factors affecting cancer in such a long term followup.

It wasn't entirely clear to me why the specific cutoffs for insulin were chosen, or why look at insulin specifically. I was wondering if there was any interaction between incidence of T2D, or fasting glycemia before or during the long followup (if available) and the incidence of cancer, as a mediating factor. I was also curious if there was an age dependent or menopause dependent effects on the outcome, obesity is considered a risk factor for breast cancer at an older age and less so (even protective?) at a younger age. Can the authors look at any interactions?

The authors point well to the stregnths and weaknesses of the study.

Congratulations on this long followup and intersting study

Any attachments provided with reviews can be seen via the following link: ********

---

* Please upload any figures associated with your paper as individual TIF or EPS files with 300dpi resolution at resubmission; please read our figure guidelines for more information on our requirements: http://journals.plos.org/plosmedicine/s/figures. While revising your submission, we strongly recommend that you use PLOS's NAAS tool (https://ngplosjournals.pagemajik.ai/artanalysis) to test your figure files. NAAS can convert your figure files to the TIFF file type and meet basic requirements (such as print size, resolution), or provide you with a report on issues that do not meet our requirements and that NAAS cannot fix.

After uploading your figures to PLOS's NAAS tool - https://ngplosjournals.pagemajik.ai/artanalysis, NAAS will process the files provided and display the results in the "Uploaded Files" section of the page as the processing is complete.

If the uploaded figures meet our requirements (or NAAS is able to fix the files to meet our requirements), the figure will be marked as "fixed" above. If NAAS is unable to fix the files, a red "failed" label will appear above.

When NAAS has confirmed that the figure files meet our requirements, please download the file via the download option, and include these NAAS processed figure files when submitting your revised manuscript.

* ETHICS STATEMENTS: Please also include an appropriate contact (web or email address) for data inquiries. Please note that this cannot be a study author.

FIGURES AND TABLES

SUPPLEMENTARY MATERIAL

REFERENCES

* Where website addresses are cited, please include the complete URL and specify the date of access (e.g. [accessed: 12/06/2025]).

STUDY TYPE-SPECIFIC REQUESTS

* Abstract: Please include the study design, population and setting, number of participants, years during which the study took place (enrollment and follow up), length of follow up, and main outcome measures.

* Please be explicit in the title and the abstract that is a post-hoc sub-study/follow-on study of the primary trial and confirm that the current study was not a pre-specified longterm outcome of the primary trial. We suggest reporting in-line with CONSORT explicitly stating the sub-study nature and ensuring that the abstract details the main trial items in 2-3 sentences, including the study population, dates, intervention and primary outcome. The majority of the abstract should then describe the complete details of this post-hoc sub-study.

* Please comment on whether the current study received ethical approval that is different from that of the primary trial.

* Please ensure that the study is reported according to the STROBE (or appropriate STOBE extension) guideline (available from: https://www.equator-network.org/reporting-guidelines/strobe) and include the completed STROBE (or STROBE extension) checklist as Supporting Information. Please add the following statement, or similar, to the Methods: "This study is reported as per the Strengthening the Reporting of Observational Studies in Epidemiology (STROBE) guideline (S1 Checklist)." When completing the checklist, please use section and paragraph numbers, rather than page numbers.

* For all observational studies, in the manuscript text, please indicate: (1) the specific hypotheses you intended to test, (2) the analytical methods by which you planned to test them, (3) the analyses you actually performed, and (4) when reported analyses differ from those that were planned, transparent explanations for differences that affect the reliability of the study's results. If a reported analysis was performed based on an interesting but unanticipated pattern in the data, please be clear that the analysis was data driven.

* Please state in the Methods section whether the study had a prospective protocol or analysis plan. If a prospective analysis plan (from your funding proposal, IRB or other ethics committee submission, study protocol, or other planning document written before analyzing the data) was used in designing the study, please include the relevant document(s) with your revised manuscript as a Supporting Information file to be published alongside your study and cite it in the Methods section. A legend for this file should be included at the end of your manuscript. If no such document exists, please make sure that the Methods section transparently describes when analyses were planned, and when/why any data-driven changes to analyses took place. Changes in the analysis, including those made in response to peer review comments, should be identified as such in the Methods section of the paper, with rationale.

---

## [Decision Letter · Decision Letter 2]

21 Nov 2025

Dear Dr. Taube,

Thank you very much for re-submitting your manuscript "A post hoc analysis of sex-specific associations between baseline insulin levels and cancer outcomes after surgery-induced weight loss: Findings from the prospective, controlled Swedish Obese Subjects study" (PMEDICINE-D-25-02306R2) for review by PLOS Medicine.

Thank you for your detailed response to the reviewers' and editors’ comments. I have discussed the paper with my colleagues, and it has also been seen again by two of the original reviewers. The changes made to the paper were mostly satisfactory to the reviewers. You will see that the statistical reviewer has a few concerns that require careful attention. As such, we intend to accept the paper for publication, pending your attention to the reviewers' and editors' comments below in a further revision. When submitting your revised paper, please once again include a detailed point-by-point response to the editorial comments. The remaining issues that need to be addressed are listed at the end of this email.

In revising the manuscript for further consideration here, please ensure you address the specific points made by each reviewer and the editors. In your rebuttal letter you should indicate your response to the reviewers' and editors' comments and the changes you have made in the manuscript. Please submit a clean version of the paper as the main article file. A version with changes marked must also be uploaded as a marked up manuscript file. Please also check the guidelines for revised papers at http://journals.plos.org/plosmedicine/s/revising-your-manuscript for any that apply to your paper.

We ask that you submit your revision by Nov 28 2025. However, if this deadline is not feasible, please contact me by email, and we can discuss a suitable alternative.

Please note that I will be out of the office from November 24 to December 5. For urgent matters, please contact polosmedicine@plos.org.

We look forward to receiving the revised manuscript.

Sincerely,

Alexandra Tosun, PhD

Senior Editor 

PLOS Medicine

plosmedicine.org

Comments from Reviewers:

Reviewer #1: The authors addressed all my concerns

Reviewer #2: I thank the authors for their considered and comprehensive responses to my comments and note the manuscript is reading very well. In all but three cases, I am happy to fully accept the authors' proposed updates and consider the items finalised. However, I do suggest the authors revisit the three items below (numbers reference my original review items). In any cases where I have suggested wording, please treat this as illustrative only i.e., I am not dictating the only acceptable wording.

3. The authors have removed the p-values from the Baseline Characteristics table as requested but have not yet updated the Methods to match. Suggest removing the first sentence of the Statistical Analysis section (line 195).

4. [MINOR] I take the authors' point on missingness but worry a reader may have the same concerns I did with the text as it stands on first read. Perhaps even something as simple as 'Due to the nature of the available data, data missingness was expected to be negligible in the primary outcomes and exposures. Where data were missing, Missing data were assumed to be "missing at random"…' This is just a wording tweak and is not required for me to recommend acceptance if the authors feel strongly.

12. Thanks to the authors for clarifying that patients in the in the main analyses were not affected by the issues raised in this point. Unfortunately, the sensitivity analysis as specified is at risk of bias. See https://link.springer.com/article/10.1007/s12561-024-09442-9 for full methodological details, but intuitively the problem can be considered as one of proportional hazards: Cox regression assumes proportional hazards but the approach as specified forces a much less hazardous first three years than follow-up afterwards. There are several ways to address this, the simplest being to include all participants whether they have an event in the first three years or not and "start the clock" at three years for all patients. That is, remove the first 3 years of person-time and all events occurring then for all participants, not just those who had early events.

Requests from Editors:

GENERAL

* Please confirm that your title complies with to PLOS Medicine's style. Your title must be nondeclarative and not a question. It should begin with main concept if possible. "Effect of" should be used only if causality can be inferred, i.e., for an RCT. Please place the study design ("A randomized controlled trial," "A retrospective study," "A modelling study," etc.) in the subtitle (ie, after a colon).

Editorial note: Based on the Abstract, we don’t think ‘baseline insulin levels’ should be in the title. We suggest: “Sex-specific associations of cancer outcomes after surgery-induced weight loss: A post hoc analysis of the Swedish Obese Subjects study"

* Statistical reporting: Please revise throughout the manuscript, including tables and figures.

- Please report statistical information as follows to improve clarity for the reader ""22% (95% CI [13,28]; p</=)"".

- Please separate upper and lower bounds with commas instead of hyphens as the latter can be confused with reporting of negative values.

- Please repeat statistical definitions (HR, CI etc.) for each set of parentheses.

* Please ensure that all abbreviations are defined at first use throughout the text (including statistical abbreviations).

* Please ensure that tables and figures, including those in supplementary files, are appropriately referenced in the main text.

* Please review your text for claims of novelty or primacy (e.g. 'for the first time' or ‘novel’) and remove this language.

* Please confirm that any use of statistical terms (such as trend or significant) are supported by the data, and if not please remove them. The term trend should be used only when the test for trend has been conducted.

* Please define all acronyms used in each figure or table in its corresponding legend.

* Please revise for use of patient-centered language. Please note that patient-centered language is constructed with the use of post-modified nouns (e.g. 'patients undergoing surgery’ (or similar) instead of ‘surgery patients’) putting the person first in the sentence structure.

* Please confirm that the metadata in the online submission form is updated and accurate.

- The Data Availability Statement (DAS) in the online submission form requires revision. Please ensure to update the statement to include the contact provided in the manuscript file.

ABSTRACT

* Please confirm that your abstract complies with our requirements, including providing all the information relevant to this study type https://journals.plos.org/plosmedicine/s/submission-guidelines#loc-abstract

* Please confirm that all numbers presented in the abstract are present and identical to numbers presented in the main manuscript text.

* In the abstract, please include the important dependent variables that are adjusted for in the analyses.

* We suggest including the original trial registration number in the Abstract Methods.

* l.49, “usual obesity care” – How is usual obesity care defined? Would ‘usual obesity-related care’ be a better description?

* “Moreover, bariatric surgery was associated with a lower overall cancer mortality rate in women (HRadj=0.78 [95% CI:0.61, 1.00]; p=0.050).” – based on the confidence intervals and the p-values, please consider re-phrasing. In the main text, you describe this result as “the association was attenuated”.

* Abstract Background: “defined by previously suggested predictors of treatment benefit.” – please provide more detail.

* Abstract Conclusions: We believe you should comment on the male-specific findings here as well.

AUTHOR SUMMARY

* In the author summary, in the final bullet point of 'What Do These Findings Mean?', please clearly state the main limitations of the study in non-technical language.

METHODS AND RESULTS

* “age >51 if data were missing” – please add ‘years’ (as a unit) when reporting age.

* Thank you for providing your STROBE checklist. Please replace the page numbers with paragraph numbers per section (e.g. "Methods, paragraph 1"), since the page numbers of the final published paper may be different from the page numbers in the current manuscript.

* Table 1: It seems that more women in the surgery group had a prior cancer diagnosis. Is ~1.5 years a considerable age difference? Please verify that you have accurately described the baseline characteristics and their differences.

* “Sensitivity analyses produced similar estimates after excluding 46 women diagnosed within the first three years (HRadj=0.82 [95% CI: 0.70,0.95]; p=0.010) and 190 with prior cancer (HRadj=0.82 [95% CI: 0.70, 0.97]; p=0.017).” – please ensure to include a reference to the relevant table/figure.

* “Among female controls, higher baseline insulin levels were associated with higher overall cancer incidence, with rates ranging from 11.6 cases per 1,000-person-years [95% CI: 9.9, 13.7] in the lowest third to 17.8 cases per 1,000-person-years [95% CI: 15.0, 21.2] in the highest third (Figure 3A).” – Please revise for clarity. Instead of describing an association, we suggest simply reporting that higher case numbers were observed in all surgery groups at all insulin levels.

* Figure 4: To avoid using the word "fatal," we suggest renaming the graphs.

* In the Methods section, please ensure that you have described how the insulin thresholds were chosen and what constitutes a normal/healthy level, citing relevant literature.

General Editorial Requests

---

## [Editor Report · Decision Letter 3]

16 Dec 2025

Dear Dr Taube, 

On behalf of my colleagues and the Academic Editor, Sanjay Basu, I am pleased to inform you that we have agreed to publish your manuscript "Sex-specific associations between surgery-induced weight loss and cancer outcomes: A post hoc analysis of the prospective, controlled Swedish Obese Subjects study" (PMEDICINE-D-25-02306R3) in PLOS Medicine. I appreciate your thorough responses to the reviewers' and editors' comments throughout the editorial process.

PRESS

Sincerely, 

Alexandra Tosun, PhD 

Senior Editor 

PLOS Medicine